# Effect of Continuous Casting and Heat Treatment Parameters on the Microstructure and Mechanical Properties of Recycled EN AW-2007 Alloy

**DOI:** 10.3390/ma17143447

**Published:** 2024-07-12

**Authors:** Grażyna Mrówka-Nowotnik, Grzegorz Boczkal, Andrzej Nowotnik

**Affiliations:** 1Department of Material Science, Rzeszow University of Technology, Al. Powstańców Warszawy 12, 35-959 Rzeszow, Poland; nowotnik@prz.edu.pl; 2Faculty of Non-Ferrous Metals, AGH University of Science and Technology, Al. Mickiewicza 30, 30-059 Cracow, Poland; gboczkal@agh.edu.pl

**Keywords:** aluminum alloys, recycling, microstructure, heat treatment, mechanical properties, LM, SEM, DSC, XRD

## Abstract

The growing use of aluminum and its compounds has increased the volume of aluminum waste. To mitigate environmental impacts and cut down on manufacturing expenses, extensive investigations have recently been undertaken to recycle aluminum compounds. This paper outlines the outcomes of a study on fabricating standard EN AW-2007 alloy using industrial and secondary scrap through continuous casting. The resultant recycled bars were analyzed for their chemical makeup and examined for microstructural features in both the cast and T4 states, undergoing mechanical property evaluations. The study identified several phases in the cast form through LM, SEM + EDS, and XRD techniques: Al_7_Cu_2_Fe, θ-Al_2_Cu, β-Mg_2_Si, Q-Al_4_Cu_2_Mg_8_Si_7_, and α-Al_15_(FeMn)_3_ (SiCu)_2,_ along with Pb particles. Most primary intermetallic precipitates such as θ-Al_2_Cu, β-Mg_2_Si, and Q-Al_4_Cu_2_Mg_8_Si_7_ dissolved into the α-Al solid solution during the solution heat treatment. In the subsequent natural aging process, the θ-Al_2_Cu phase predominantly emerged as a finely dispersed hardening phase. The peak hardness achieved in the EN AW-2007 alloy was 124.8 HB, following a solution heat treatment at 500 °C and aging at 25 °C for 80 h. The static tensile test assessed the mechanical and ductility properties of the EN AW-2007 alloy in both the cast and T4 heat-treated states. Superior strength parameters were achieved after solution heat treatment at 500 °C for 6 h, followed by water quenching and natural aging at 25 °C/9 h, with a tensile strength of 435.0 MPa, a yield strength of 240.5 MPa, and an appreciable elongation of 18.1% at break. The findings demonstrate the feasibility of producing defect-free EN AW-2007 alloy ingots with excellent mechanical properties from recycled scrap using the continuous casting technique.

## 1. Introduction

Aluminum alloys have become crucial non-ferrous metals in contemporary technology due to their abundance in the earth’s crust and superior properties, including low density, high strength-to-weight ratio, excellent thermal and electrical conductivity, corrosion resistance, and advantageous technological traits [1,2,3,4,5,6,7,8]. These metals are particularly valued because they can be recycled repeatedly without property degradation, consuming 95% less energy than primary aluminum production [9,10,11,12,13,14,15,16,17]. As a result, about 75% of all aluminum ever produced is still in use, showcasing its sustainability. The energy consumption difference is notable between primary aluminum production, around 45 kWh/kg, and secondary production, about 2.8 kWh/kg, leading many manufacturers to prefer recycled aluminum [9,10,11,12,13]. From an environmental standpoint, enhancing the utilization of recycled metal is vital, as the production of aluminum through recycling emits merely around 4% of the CO_2_ compared to that of primary production [11,18,19,20,21,22,23].

Research has increasingly focused on the recycling of aluminum alloys, especially the 2xxx series, which includes Al-Cu-Mg-(Si) alloys. These alloys are essential for high-load applications in aircraft structures, automotive parts, rolling stock, and construction due to their high strength, fracture resistance, fatigue properties, and damage tolerance [1,2,3,4,5,6,7,8,9,24,25,26,27]. The performance of the 2xxx series alloys hinges on their chemical composition and specific heat treatments [1,2,3,4,5,6,7,24,25,26,27,28]. They include the main alloying elements such as Cu, Mg, and Si, alongside transition metals like Mn, Fe, Cr, Ni, and Ti. The EN AW-2007 alloy, also part of this series, contains lead and is noted for its high strength, excellent machinability, and moderate corrosion resistance [29]. This makes the EN AW-2007 alloy popular in various demanding applications, including

Aerospace components like fittings, fasteners, and brackets;Automotive parts, such as engine and transmission components;Military applications requiring precision-machined parts;Industrial machinery components like gears and shafts.

The 2xxx group alloys, like many technical aluminum alloys, possess a multiphase microstructure. The primary phase components in these alloys, apart from the α-Al solution (enriched with copper and magnesium), are precipitates from complex intermetallic phases. These intermetallics typically arise from the interaction between aluminum and primary alloying elements like Cu and Mg, combined with transition metals such as Fe, Mn, Cr, Ni, and Ti. Some precipitates also originate from elements like Fe and Si, which are difficult to eliminate during the aluminum metallurgical process and have low solubility in solid aluminum. Even trace amounts of these elements can lead to the formation of intermetallic phases, which are generally brittle and prone to fracture during plastic deformation. These phases also sequester some primary alloying elements, restricting their contribution to precipitation hardening [3,6,7,11,30]. Based on our investigations [13,26] and references in the literature [29,30,31,32,33,34,35,36,37,38,39,40], observed intermetallic phases in 2xxx alloys include binary θ-Al_2_Cu, β-Mg_2_Si; ternary S-Al_2_CuMg, β-Al_7_Cu_2_Fe, Al_6_(FeCu); quaternary phases Q-Al_4_CuMgSi_4_, α-Al_12_(FeMn)_3_Si; and a five-component phase α-Al_15_(CuFeMn)_3_Si_2_. Additionally, in alloy 2007, precipitates composed of Pb are present [30,31,32,33,34,35,36,37,38,39,40].

The alloy’s phase composition and the morphology of microstructural components can be significantly altered through heat treatment processes, profoundly affecting mechanical properties. The formation of secondary intermetallic phases during the aging of solution heat-treated alloys is crucial for precipitation hardening, markedly enhancing the mechanical properties of 2xxx series aluminum alloys. The type and volume fraction of these strengthening phases primarily depend on the alloy’s chemical composition and the proportion of elements such as Cu, Mg, and Si. Key strengthening phases include θ-Al_2_Cu, β-Mg_2_Si, S-Al_2_CuMg, and quaternary complexes such as Q-(Cu_2_Mg_8_Si_6_Al_4_, Al_5_Cu_2_Mg_9_Si_7_, or Al_4_Cu_2_Mg_8_Si_7_) [34,35,36,37,38,39,40]. The specific alloying proportions, particularly the Cu/Mg and Mg/Si ratios, determine which of these phases strengthen the alloy most effectively. The Cu/Mg ratio affects the precipitation of the θ and S phases from solution heat-treated alloys, whereas the addition of Si modifies the precipitation sequence, promoting the formation of β (Mg_2_Si) and Q (Cu_2_Mg_8_Si_6_Al_4_) phases along with θ (Al_2_Cu). With a high Cu content and an Mg/Si > 1 ratio, the β phase may form alongside the θ phase. Conversely, with an Mg/Si < 1 ratio, either the Q or S phase may develop depending on Si levels. A very low Si content favors the S phase, whereas a higher Si content encourages the formation of the Q phase [34,35,36,37,38,39,40]. 

In this research, the precipitation sequence of strengthening phases from the solution heat-treated recycled alloy was determined using differential scanning calorimetry (DSC) and X-ray diffraction (XRD) tests. The study also examined how different heat treatment conditions affect the mechanical properties of the alloy, implementing various solution heat treatment temperatures and natural aging at approximately 25 °C. The mechanical characteristics of the 2007 alloy, obtained from recycled scrap, were assessed by conducting hardness measurements and static tensile testing. The optimal solution heat treatment and aging conditions were identified to maximize the alloy’s mechanical performance while maintaining its ductility.

Understanding the heat treatment processes is vital for optimizing the mechanical properties of 2xxx series aluminum alloys. These processes, which include solution heat treatment, quenching, and artificial or natural aging, play a decisive role in manipulating the microstructure of these alloys. During heat treatment, the precipitates that form within the alloy structure, such as θ-Al_2_Cu and β-Mg_2_Si, significantly enhance the alloy’s strength and durability through precipitation hardening. However, achieving the desired properties requires precise control over the temperature and duration of each process step, as well as over the cooling rates during quenching. Current heat treatment practices face several challenges, particularly in ensuring uniform properties throughout the alloy. Mismanagement of heat treatment parameters can lead to the development of non-uniform microstructures, which adversely affect the mechanical properties and can introduce stresses that lead to premature failure under operational loads. Moreover, variations in alloying elements can lead to the formation of undesirable phases that reduce the effectiveness of precipitation hardening. Addressing these issues is vital for advancing the application of 2xxx series aluminum alloys in critical sectors, where enhanced performance and reliability are required.

## 2. Materials and Methods

The material investigated in this study was an EN AW-2007 aluminum alloy, created from recycled sources. Ingots were produced using a multi-strand continuous casting machine at Lukasiewicz Foundry Research Institute, Cracow (Table 1). The casting process involved four oil-lubricated crystallizers (Figure 1). The starting materials included a significant amount of scrap and manufacturing residues, such as chips from the EN AW-2007 alloy. Around 250 kg of this scrap was melted down in a crucible resistance furnace. Before forming the molten alloy into ingots, routine samples were taken to check the chemical composition. Additional alloying elements were incorporated as needed to fulfill the specifications of the PN EN 573-1 standard [41]. The molten alloy underwent refinement through argon gas barbotage, using refinement settings of 10 min and a gas flow of 10 L per minute. After stabilizing and reaching the required composition, ingot casting was initiated according to the specified parameters. The chemical composition of the ingots was analyzed using a ARL 3460 spectrometer, Thermo, Lausanne, Switzerland.

### Microscopic Examination

The microstructural analysis of the EN AW-2007 alloy in both the as-cast and heat-treated conditions was carried out using light microscopes, along with a scanning electron microscope equipped with an EDS. Samples, sized at 2.5 cm^2^, were extracted using a precision cutting machine and embedded in bakelite, ground with SiC papers, polished with diamond suspensions, and observed under both etched (using modified Keller’s reagent) and non-etched conditions. Fractographic analyses on samples from static tensile tests were also performed.

Quantitative microstructural analysis post-casting, solution heat treatment, and natural aging of the 2007 alloy was conducted with an X-ray diffractometer, using solid, polished samples and a filtered copper lamp, to identify phases based on the Powder Diffraction File by the ICDD [42]. DSC testing was executed on cast and solution heat-treated specimens to record thermal effects related to the separation of strengthening phases, conducted under a protective argon atmosphere to avoid oxidation. Samples for DSC were prepared in disc shapes, 3 mm in diameter and 1 mm in thickness, weighing about 30 ± 0.1 mg. These were heated continuously at a rate of 10 °C/min from a starting temperature of 25 °C up to 700 °C.

The precipitation strengthening process during heat treatment was based on calorimetric findings, with samples heated to form a homogeneous solid α-Al solution (held for 6 h), then quenched in water at about 15–20 °C and naturally aged to reach the T4 state. Mechanical properties were evaluated through hardness testing using a Brinell hardness tester and a static tensile test following the PN-EN 10002-1:2004 standard [43]. Tensile test specimens had a diameter of 10 mm (Figure 2). Tests were performed on samples derived from scrap and subjected to solution heat treatment and natural aging, with measurements on hardness and tensile strength including on samples conditioned at various temperatures and aging durations (ranging from 0.5 to 200 h), to determine yield strength, tensile strength, and elongation. The elongation measurements were accurately recorded using an extensometer, in compliance with ASTM E8/E8M [44]. This standard prescribes detailed procedures for conducting tensile tests, including the use of an extensometer to ensure precise and reliable measurements of material ductility.

## 3. Results and Discussion

In the continuous casting process, four rod-shaped ingots were produced, each measuring 70 mm in diameter and 5000 mm in length. The outer surfaces of these ingots were free from defects like cracks, blow holes, fusions, and swells (Figure 3a). Samples were taken from each ingot for initial spectrometric testing to verify the alloy’s chemical composition, confirming that the content of alloying elements met the specifications set out for the 2007 alloy standard [41] (Table 2).

The macrostructure and microstructure of the EN AW-2007 alloy ingots were thoroughly examined, revealing no discontinuities or casting flaws. The ingots demonstrated a uniform and fine-grained structure across their entire cross-section, which suggests that the alloy should possess strong mechanical properties post-casting. Produced from scrap via continuous casting, these EN AW-2007 alloy ingots exhibited a hardness of 84 HB. Mechanical properties assessed through static tensile testing included a tensile strength (R_m_) of 286.8 MPa, yield strength (R_0.2_) of 160.0 MPa, and relative elongation (A_5_) of 8.2%

The microstructure observed in a sample from the EN AW-2007 alloy ingot, characteristic of the cast condition, includes α-Al solid solution dendrites with intermetallic phase precipitates. These precipitates, primarily eutectic in nature, are distributed within the interdendritic spaces (Figure 4). The principal alloying elements in the 2xxx series, specifically Cu, Mg, and Si, are partially dissolved in the solid solution, enhancing its strength via solution strengthening. The other elements combine to form primary intermetallic phases with aluminum and transition metals (e.g., Fe and Mn). Spheroidal-shaped precipitates were notably present in the microstructure (Figure 4b).

SEM imaging along with EDS microanalysis (Figure 5 and Table 3) and XRD diffraction analysis (Figure 6) were used to define the phase components within the alloy’s microstructure. Detailed SEM observations revealed that the primary precipitates formed during solidification displayed diverse shapes and colors, such as “Chinese script”, spheroids, plaques, needles, and polyhedrons (Figure 4 and Figure 5). The phase composition of the recycled alloy post-casting was found to include Pb particles and precipitates of phases like θ-Al_2_Cu and β-Mg_2_Si, along with Al_7_Cu_2_Fe, Q-Al_4_Cu_2_Mg_8_Si_7_, and α-Al_15_(FeMn)_3_(SiCu)_2_ phases.

Microscopic examination and XRD spectral analysis reveal that apart from the α-Al solid solution, the binary θ-Al2Cu phase precipitates constitute the most substantial volume fraction. The XRD spectrum of the alloy post-casting featured numerous high amounts of signals characteristic of this phase (Figure 6).

The calorimetric analysis played a vital role in determining the optimal annealing temperature for the supersaturation of the 2007 alloy (Figure 7a). Two prominent endothermic peaks were identified: the first peaking at 507 °C, attributed to the dissolution of eutectic intermetallic phases, and the second at 628 °C, associated with the dissolution of the α-Al solid solution. A detailed examination of the DSC curve around the first peak enabled precise determination of the annealing temperature, ensuring that the scrap-derived alloy would not overheat during the annealing process. The chosen temperatures for supersaturation were 490, 500, and 510 °C (Figure 7a), with the heat treatment protocol outlined to achieve the T4 condition of the alloy shown in Figure 7b.

The mechanical properties of the EN AW-2007 alloy post-casting can be significantly improved through precipitation hardening. Microscopic examinations (Figure 8), hardness testing (Figure 9), and XRD analyses (Figure 10) show how solution treatment affects the microstructure and hardness. Scanning electron microscopic analyses of samples after solution heat treatment at 490, 500, and 510 °C for 6 h, followed by water quenching, showed that most primary intermetallic precipitates observed post-casting (Figure 4, Figure 5, Figure 6, Figure 8a and Figure 10a,b) dissolved into the α-Al solid solution during the annealing period (Figure 8b–d and Figure 10c,d). Pb particles did not dissolve and kept their spheroidal shape (Figure 8b–d and Figure 10c,d), as seen in the cast state (Figure 7a). The structure of the remaining non-dissolved intermetallic precipitates changed, mainly in those containing Fe, with lamellar precipitates of the Al_7_Cu2Fe phase partly changing into more compact forms (Figure 8) and transforming into α-Al_15_(FeMn)_3_(SiCu)_2_ phase particles. The sample heated to the highest temperature of 510 °C (Figure 8d) had a lower volume of primary intermetallic precipitates than the sample treated at 490 °C (Figure 8b). Using higher annealing temperatures increases the solubility of primary intermetallic particles like θ-Al_2_Cu, β-Mg_2_Si, and Q-Al_4_Cu_2_Mg_8_Si_7_ in the α-Al solution. Hardness tests on samples annealed at 490, 500, and 510 °C and then supersaturated showed that the annealing temperature affects the alloy’s hardness after cooling (Figure 9).

The hardness values observed across different supersaturation temperatures for the EN AW-2007 alloy are relatively stable within the range examined. Specifically, at 500 °C, the alloy exhibited a hardness of 90.2 HB; at 510 °C, the hardness slightly decreased to 89.0 HB; and at 490 °C, it was recorded at 87.8 HB. These minor differences suggest that the hardness of the alloy remains relatively consistent, regardless of small variations in the supersaturation temperature. This consistency in hardness, particularly at 500 °C and 510 °C, can be attributed to the almost complete dissolution of intermetallic phases into the solid solution, as illustrated in Figure 8b–d and Figure 10c,d. This dissolution results in the homogenization and strengthening of the solution heat-treated EN AW-2007 alloy, maintaining a stable level of hardness typical for this temperature range. The treatment process, therefore, does not significantly impact the hardness, which is preserved due to the characteristics of the alloy composition and the thermal conditions applied. After supersaturation, the samples underwent natural aging for 200 h. Throughout this aging period, the precipitated phase particles dispersed evenly throughout the alloy, enhancing its structural integrity. Diffractometric studies indicate that this strengthening is predominantly due to the precipitation of Al_2_Cu phases (Figure 10e,f).

Natural aging further enhanced the alloy’s hardness. After solution heat treatment, samples were naturally aged at approximately 25 °C for 200 h, reaching the T4 condition. Hardness measurements taken after various natural aging durations were used to generate aging plots of HB vs. time (Figure 11). These curves show that the alloy’s hardness peaks after roughly 20 h of aging, independent of the supersaturation temperature, and after a slight decrease, it rises again to a second peak on the curve, marking the highest hardness of the alloy. The impact of the supersaturation temperature was evident, as after peaking at 121.6 HB at 490 °C/60 h and 119.4 HB at 510 °C/45 h, the hardness values dropped to about 105–107 HB and remained stable up to 200 h of aging. The maximum hardness observed was 124.8 HB in the alloy supersaturated at 500 °C and naturally aged for 80 h (Figure 11a). According to the calorimetric curve, during continuous heating of the treated alloy (Figure 11b), the precipitation process starts with Cu atom clusters forming and progresses with the creation of GP zones coherent with the matrix. The most intense exothermic peaks come from the precipitation of θ” and θ’ phases, which are partially coherent with the matrix, optimizing the hardness and mechanical properties of the alloy. Continued heating leads to the dissolution of metastable phases and the formation of stable equilibrium precipitates of θ-Al_2_Cu phases, incoherent with the matrix (Figure 11b).

Analysis of the microstructural images of the alloy aged at 25 °C/25 h and 200 h confirms the hardness measurements and calorimetrical analysis results (Figure 11). After 25 h of natural aging, the microstructure of the 2007 alloy shows undissolved primary intermetallic phase particles and a few very fine spheroidal secondary intermetallic phase particles uniformly distributed in the α-Al solid solution (Figure 12a). Extending the aging time to 200 h increases the relative volume of strengthening phases in the alloy’s microstructure (Figure 12b). This corresponds to the phase of decreasing hardness on the aging curve (Figure 11a), indicating overaging, which demonstrates the growth and coagulation of the secondary phase strengthening particles (Figure 12b).

A series of static tensile tests were conducted on samples subjected to supersaturation followed by natural aging for various periods. Table 4 and Figure 13 illustrate how the natural aging duration influences the mechanical properties of the recycled EN AW-2007 alloy.

The analysis of the results indicates that the yield strength (R_0.2_) increases with the duration of aging time, regardless of the solution heat treatment temperature, with samples treated at 500 °C achieving higher R_0.2_ values (Figure 13a). However, the R_m_ value of the alloy treated at 500 °C peaks at 435 MPa after 9 h of natural aging (Figure 13b). At this aging duration, the alloy also achieves the highest plasticity (Figure 13c). An alloy supersaturated at 510 °C shows similar trends but reaches lower R_m_ values. Extended natural aging results in a decline in both tensile strength (R_m_) and plastic properties. For the alloy treated at 490 °C, both the conventional yield strength (R_0.2_) and tensile strength (R_m_) increase with aging time, with the highest values attained after 80 h of aging. Nonetheless, the optimal combination of strength and plasticity was achieved for samples treated at 500 °C.

The alloys produced in this study exhibit higher mechanical properties in the as-cast state than those of the EN AW-2007 alloy reported in the literature [45,46,47]. The alloy in the T4 condition shows a yield strength (R_0.2_) similar to the values presented in data in the literature. Meanwhile, the alloy obtained through the continuous casting process using chips in the T4 condition is characterized by a higher ultimate tensile strength (R_m_) compared to the alloy in the same condition obtained through the metallurgical process using traditional manufacturing methods (Table 5) [45,46,47].

Microstructural observations of the fracture surfaces from the static tensile test on samples subjected to different solution heat treatment temperatures and natural aging times revealed features typical of ductile fracture, such as pits and voids (Figure 14). Regardless of the applied heat treatment parameters, intergranular cracking predominated in the EN AW-2007 alloy, with cracks propagating through nucleation, growth, and void coalescence. Secondary hardening phase θ-Al_2_Cu and Q-Al_4_Cu_2_Mg_8_Si_7_ particles, along with undissolved primary intermetallic phase θ-Al_2_Cu, Al_7_Cu_2_Fe precipitates, α-Al_15_(FeMn)_3_(SiCu)_2_, and lead precipitates, significantly influenced the cracking process (Figure 14e,f). The primary sites for void nucleation were the secondary precipitates of the θ-Al_2_Cu and Q-Al_4_Cu_2_Mg_8_Si_7_ hardening phases. Additionally, in areas with undissolved primary θ-Al_2_Cu precipitates, decohesion initiated at the matrix–particle interface or due to tensile stresses, leading to the cracking of hard, brittle primary phase particles containing iron, such as α-Al_15_(FeMn)_3_(SiCu)_2_ (Figure 14e).

## 4. Conclusions

The findings presented in this article confirm the feasibility of producing EN AW-2007 alloy ingots from scrap via continuous casting. The observed macrostructure and microstructure of the ingots showed no discontinuities or casting defects and were characterized by a uniform and fine-grained structure throughout the cross-section. The ingots displayed good mechanical properties in the as-cast state, with a tensile strength (R_m_) of 286.8 MPa, yield strength (R_0.2_) of 160.0 MPa, hardness of 84 HB, and relative elongation (A_5_) of 8.2%.The microstructure of the EN AW-2007 alloy ingot comprises α-Al solid solution dendrites and precipitates of intermetallic phases, including two-component θ-Al_2_Cu and β-Mg_2_Si phases, a three-component Al_7_Cu_2_Fe phase, a four-component Q-Al_4_Cu_2_Mg_8_Si_7_ phase, a five-component α-Al_15_(FeMn)_3_(SiCu)_2_ phase, and Pb particles. During annealing for supersaturation, primary particles of intermetallic phases, particularly θ-Al_2_Cu, Q-Al_4_Cu_2_Mg_8_Si_7,_ and β-Mg_2_Si, dissolved into the α-Al solid solution. However, Pb particles remained undissolved, retaining their spheroidal shape as in the cast state.The heat treatment parameters that achieved the T4 strengthening state for the EN AW-2007 cast alloy, yielding the highest hardness of 124.8 HB, included annealing at 500 °C for 6 h, supersaturation in cold water, and natural aging for 80 h. Conversely, the highest tensile strength (R_m_) of 435 MPa, along with the best plasticity (A_5_ = 18.1%), were attained after only 9 h of natural aging.Fractographic studies revealed that regardless of the heat treatment parameters, decohesion under tensile stress in the studied alloy occurs through the nucleation, growth, and coalescence of voids. Additionally, it was observed that in areas containing primary undissolved Pb particles and θ-Al_2_Cu phases, the decohesion process initiated at the matrix–particle interface. Particles of primary, hard, and brittle phases that contain Fe, such as α-Al_15_(FeMn)_3_(SiCu)_2_ and Al_7_Cu_2_Fe, were found to fragment under tensile loading.

## Figures and Tables

**Figure 1 materials-17-03447-f001:**
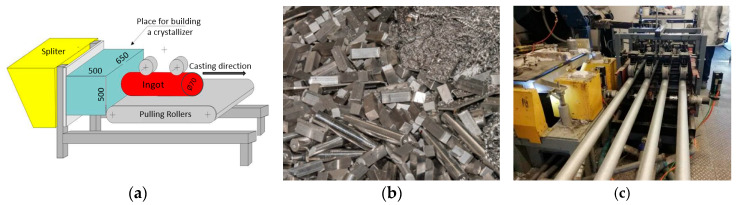
Diagram of the continuous casting setup (**a**). The scrap material used as the foundation for melting and producing the technical EN AW-2007 alloy (**b**). The procedure for casting ingots with a diameter of 70 mm using a four-rod system (**c**).

**Figure 2 materials-17-03447-f002:**
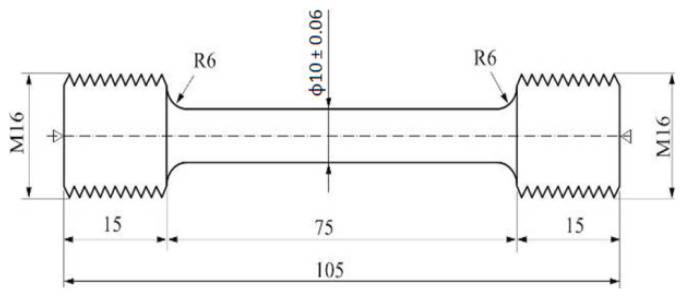
Shape and dimensions of samples used in the static tensile test.

**Figure 3 materials-17-03447-f003:**
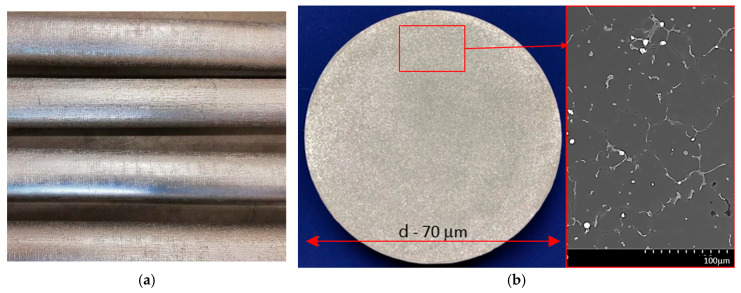
Images showing the acquired EN AW-2007 aluminum alloy ingots (**a**) and the standard macro- and microstructure across the ingot’s cross-section (**b**).

**Figure 4 materials-17-03447-f004:**
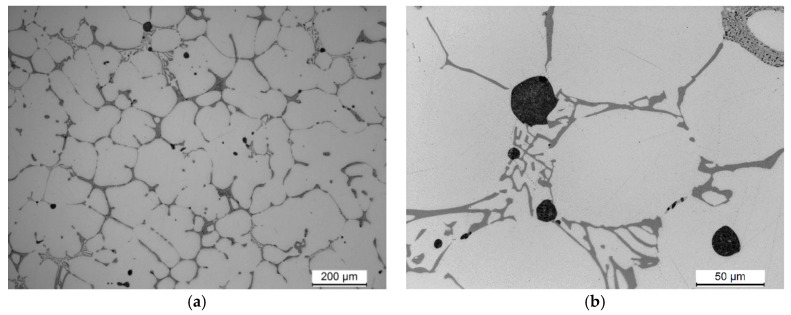
Microstructure of alloy derived from recycled materials following the casting process (**a**) mag. 10×, (**b**) mag. 100×.

**Figure 5 materials-17-03447-f005:**
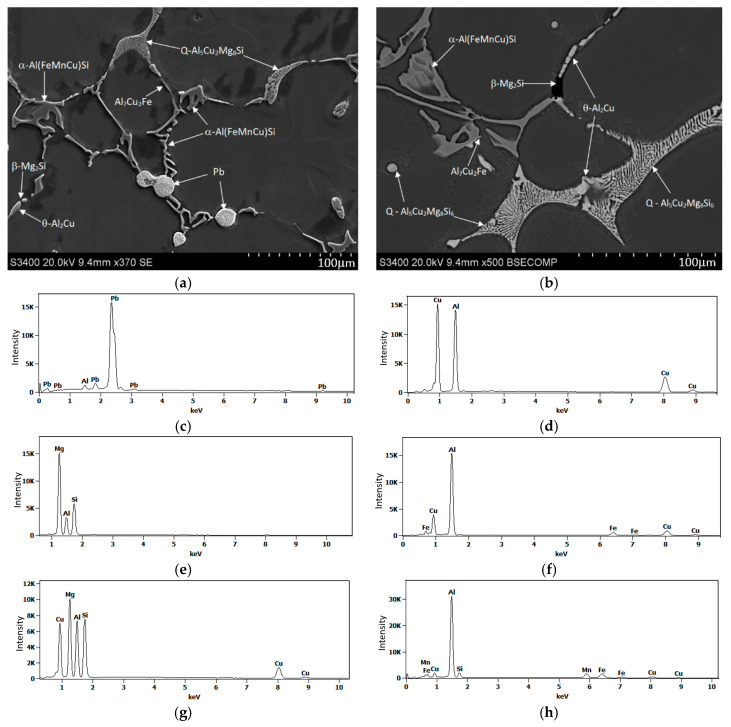
SEM micrographs of the recycled alloy after casting (**a**,**b**) display the microstructure, and typical EDS spectra for the analyzed particles are shown: (**c**) Pb, (**d**) θ-Al_2_Cu, (**e**) β-Mg_2_Si, (**f**) Al_7_Cu_2_Fe, (**g**) Q-Al_4_Cu_2_Mg_8_Si_7_, and (**h**) α-Al_15_(FeMn)_3_(SiCu)_2_ phases.

**Figure 6 materials-17-03447-f006:**
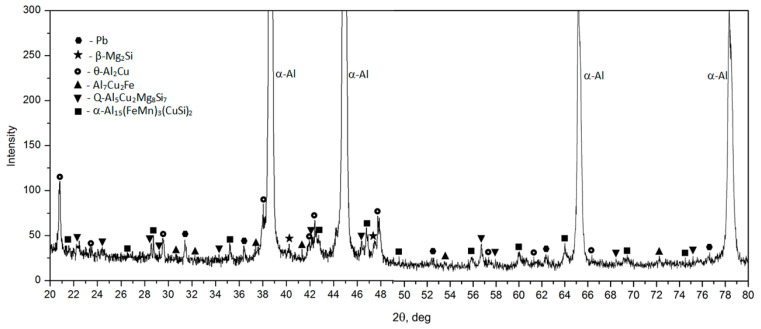
X-ray diffraction pattern of the EN AW-2007 alloy after casting.

**Figure 7 materials-17-03447-f007:**
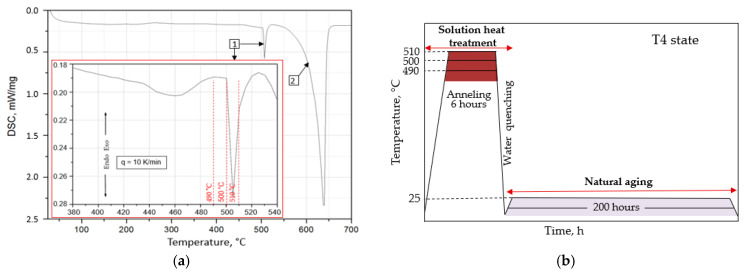
Differential scanning calorimetry curve captured while heating the EN AW-2007 alloy from ambient temperature to 700 °C at a 10 °C/min rate, featuring an expanded view of the temperature interval from 360 °C to 540 °C (**a**). Diagram of the heat treatment process (**b**).

**Figure 8 materials-17-03447-f008:**
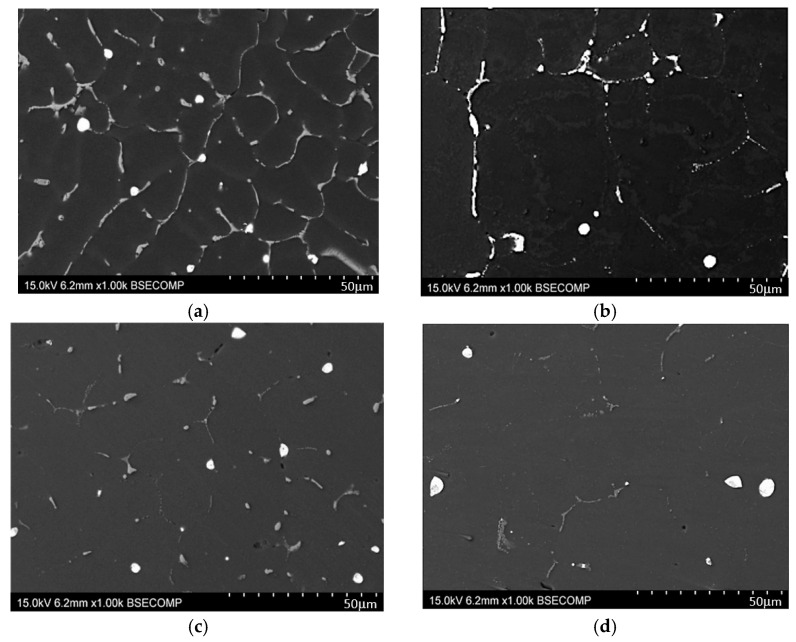
Microstructure examples of the EN AW-2007 alloy (**a**) in the as-cast condition and following solution heat treatment at (**b**) 490 °C; (**c**) 500 °C; (**d**) 510 °C.

**Figure 9 materials-17-03447-f009:**
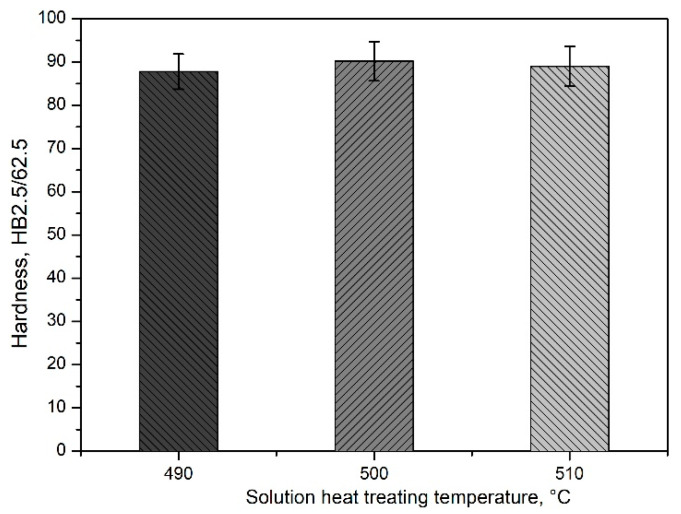
Effect of solution heat treatment temperature on the hardness of the EN AW-2007 alloy.

**Figure 10 materials-17-03447-f010:**
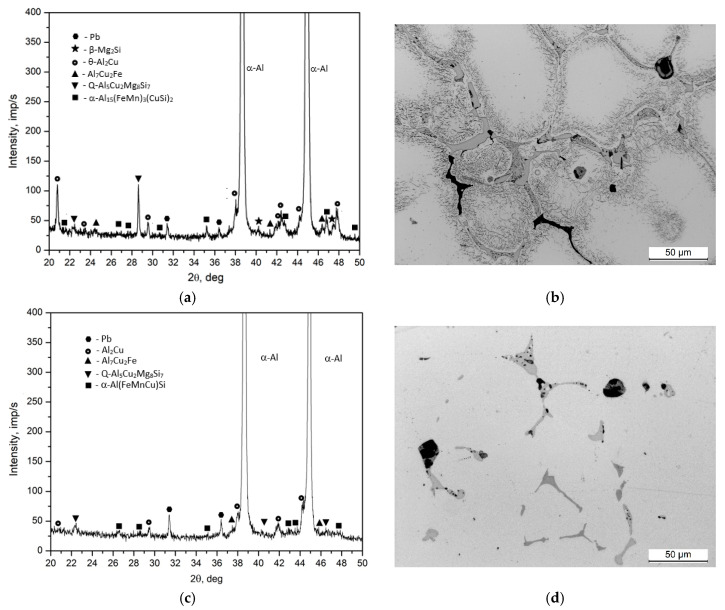
XRD profiles and microstructure of EN AW-2007 alloy in various conditions: (**a**,**b**) as-cast, (**c**,**d**) after solution treatment, and (**e**,**f**) following solution treatment and natural aging at about 25 °C for 200 h.

**Figure 11 materials-17-03447-f011:**
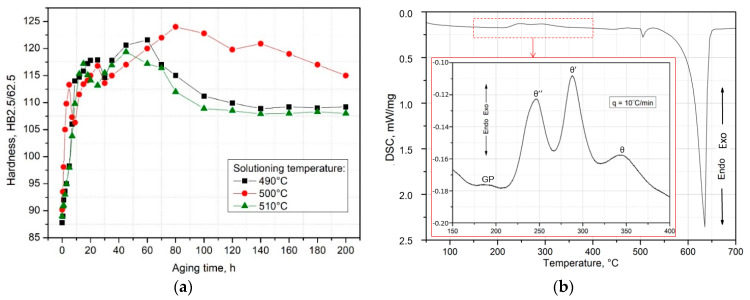
Influence of solutionizing temperature on the hardness variation of a naturally aged alloy (**a**). Differential scanning calorimetric diagram obtained during heating at a scanning rate of 10 °C/min for a solution-treated sample (**b**).

**Figure 12 materials-17-03447-f012:**
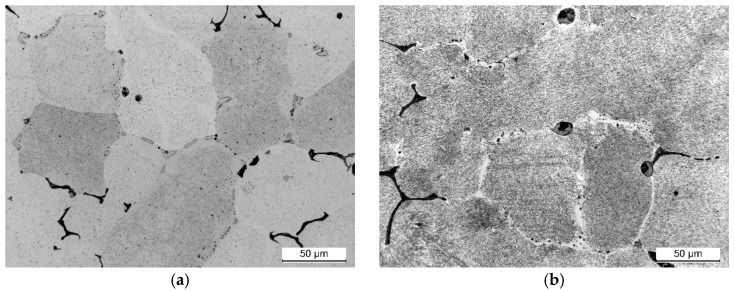
LM images of EN AW-2007 solutionized and heat-treated at 25 °C for (**a**) 25 h and (**b**) 200 h.

**Figure 13 materials-17-03447-f013:**
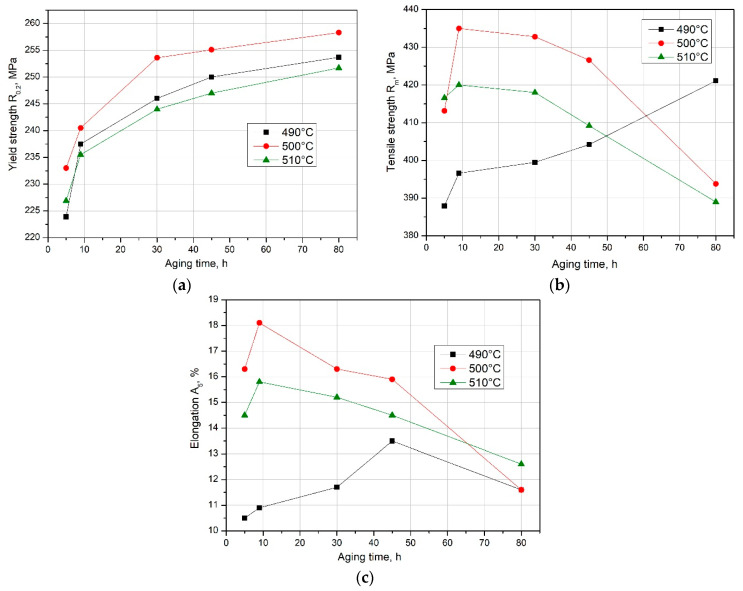
Effect of solution heat treatment temperature and natural aging duration on (**a**) yield strength R_0.2_, (**b**) tensile strength R_m_, and (**c**) relative elongation A_5_ of the EN AW-2007 alloy.

**Figure 14 materials-17-03447-f014:**
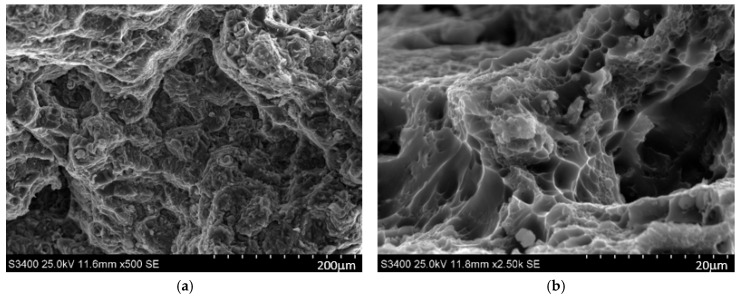
Microstructure of fracture surfaces of samples heat-treated at 490 °C for 30 h (**a**,**b**) and at 500 °C for 30 h (**c**,**d**), following a static tensile test, showing primary precipitate particles of the θ-Al_2_Cu and α-Al_15_(FeMn)_3_(SiCu)_2_ phases (**e**) and fragmented precipitates of θ-Al_2_Cu and Pb caused by tensile stress during the test (**f**).

**Table 1 materials-17-03447-t001:** Parameters for continuous casting of EN AW-2007 alloy.

Parameters	Value
Metal temperature in furnace	730 °C
Metal temperature in classifier	680–700 °C
Cooling water volume	25 L/min
Casting speed	3.5–4.0 mm/s
Remarks	Total amount of water for four crystallizers 120 L/m

**Table 2 materials-17-03447-t002:** Chemical composition of EN AW-2007 (Al-balance) wt %.

Alloy	Elements Content, wt %									
	Si	Fe	Cu	Mn	Mg	Cr	Ni	Zn	Pb	Ti
EN AW-2007	0.76	0.45	3.43	0.64	0.70	0.023	0.008	0.20	0.53	0.027

**Table 3 materials-17-03447-t003:** Chemical composition of intermetallic phase particles (determined using SEM/EDS) in the as-cast EN AW-2007 alloy.

Phase	%	Elements						
		Al	Si	Mn	Fe	Mg	Cu	Pb
Pb	wt	1.3 ÷ 1.8	-	-	-	-	-	98.2 ÷ 98.7
	at	9.0 ÷ 4	-	-	-	-	-	90.6 ÷ 91.0
θ-Al_2_Cu	wt	46.9 ÷ 47.6	-	-	-	-	52.4 ÷ 53.1	-
	at	67.1 ÷ 68.5	-	-	-	-	31.9 ÷ 32.6	-
β-Mg_2_Si	wt	2.45 ÷ 3.8	52.6 ÷ 61.2	-	-	37.9 ÷ 48.2	-	-
	at	2.55 ÷ 3.7	47.2 ÷ 48.9	-	-	48.5 ÷ 52.9	-	-
Al_7_Cu_2_Fe	wt	61.4 ÷ 65.1	-	-	8.8 ÷ 9.3	-	25.1 ÷ 29.0	
	at	75.6 ÷ 78.7	-	-	5.4 ÷ 5.7	-	13.2 ÷ 14.7	
Q Al_5_Cu_2_Mg_8_Si_6_	wt	33.7 ÷ 56.4	16.5 ÷ 26.9	-	-	15.5 ÷ 23.5	11.5 ÷ 18.7	-
	at	37.1 ÷ 59.6	16.5 ÷ 27.6		-	18.5 ÷ 28.1	5.2 ÷ 8.7	-
α-Al_15_(FeMn)_3_(SiCu)_2_	wt	56.7 ÷ 59.91	5.7 ÷ 6.8	9.6 ÷ 12.3	17.4 ÷ 19.6	-	4.2 ÷ 7.3	-
	at	71.0 ÷ 72.9	7.0 ÷ 8.1	5.7 ÷ 7.2	10.2 ÷ 11.7	-	2.6 ÷ 3.7	-

**Table 4 materials-17-03447-t004:** Influence of solution heat treatment temperature and natural aging duration on the mechanical properties of EN AW-2007 alloy.

Aging Time, h	Solution Temperature, °C								
	490			500			510		
	R_0_._2_, MPa	R_m_, MPa	A_5_, %	R_0_._2_, MPa	R_m_, MPa	A_5_, %	R_0_._2_, MPa	R_m_, MPa	A_5_, %
5	223.9 ± 2.1	387.9 ± 1.7	10.5 ± 0.2	233.0 ± 1.1	413.1 ± 1.1	16.3 ± 0.1	226.9 ± 1.8	416.6 ± 1.8	14.5 ± 0.1
9	237.5 ± 2.2	396.6 ± 2.1	10.9 ± 0.1	240.5 ± 2.7	435.0 ± 1.3	18.1 ± 0.1	235.5 ± 2.2	420.0 ± 1.3	15.8 ± 0.3
30	246.0 ± 1.6	399.5 ± 1.9	11.7 ± 0.2	253.6 ± 1.3	432.8 ± 1.4	16.3 ± 0.1	244.0 ± 1.6	418.0 ± 0.9	15.2 ± 0.4
45	250.0 ± 3.2	404.2 ± 2.3	13.5 ± 0.1	255.1 ± 1.4	426.6 ± 1.8	15.9 ± 0.1	247.0 ± 1.2	409.2 ± 2.1	14.5 ± 0.2
80	253.7 ± 0.9	421.1 ± 2.2	11.6 ± 0.3	258.3 ± 2.1	393.8 ± 2.2	11.6 ± 0.1	251.7 ± 1.9	389.0 ± 2.5	12.6 ± 0.7

The values in the table include the measurement error for each parameter. The measurement error indicates the possible range within which the true value of the measurement lies.

**Table 5 materials-17-03447-t005:** Comparison of the mechanical properties of the EN AW-2007 alloy made from scrap presented in this study with results for alloys from the literature data [45,46,47].

Alloy Condition	R_0.2_, MPa	R_m_, MPa	A_5_, %	Reference
As-cast	81	184	21	[45]
	160	286.8	8.2	*
T4	220	340	8	[45]
	240	380	8	[46]
	210–250	330–370	7–8	[47]
	240.5	435	18.1	*

* Mechanical properties of the EN AW-2007 alloy made from scrap presented in this study.

## Data Availability

The original contributions presented in the study are included in the article, further inquiries can be directed to the corresponding author.

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
