# Peer review of "Effect of Continuous Casting and Heat Treatment Parameters on the Microstructure and Mechanical Properties of Recycled EN AW-2007 Alloy"

_materials, 2024, doi:10.3390/ma17143447_

Round 1
Reviewer 1 Report
Comments and Suggestions for Authors
1. Figures 1b and 1c do not match the icons.
2. The sentence of “…yield strength of 268.7 MPa…”in line 24 is not match the data in Table 4.
3. The manuscript mainly focuses on the influence of heat treatment on the microstructure and properties of recycled EN2007 alloy. It is suggested to modify the title of the paper to be more appropriate.
4. The introduction does not provide a good introduction to the impact of heat treatment process on the microstructure and properties of 2XXX alloy, and does not indicate the issues of present processes.
5. The sentence of “However, Rm value of the alloy treated at 500 °C peaks at 435 MPa after 5 hours of natural aging (Figure 13b).” in line 299 is not match to the data in Fig.13b.
6. The manuscript lacks of necessary discussion on research results
7. The manuscript lacks necessary comparison in the mechanical properties of alloys, such as other research results or standards.
8. Carefully check the accuracy of the data in the paper.
Comments on the Quality of English LanguageCarefully check the language in the paper.
Author Response
Comments 1: Figures 1b and 1c do not match the icons.
Response 1: Thank you for this remark, indeed drawings b and c were swapped and the caption did not match the drawing. The drawings have been replaced
Comments 2: The sentence of “…yield strength of 268.7 MPa…”in line 24 is not match the data in Table 4.
Response 2: In the sentence "...yield strength of 268.7 MPa..." there was an error in line 24. The correct value of the yield strength is 240.5 MPa. It was corrected in the text
Comments 3: The manuscript mainly focuses on the influence of heat treatment on the microstructure and properties of recycled EN2007 alloy. It is suggested to modify the title of the paper to be more appropriate.
Response 3: Perhaps the following title would be more appropriate and in keeping with the content:
“Effect of continuous casting and heat treatment parameters on the microstructure and mechanical properties of recycled EN2007 alloy“
Comments 4: The introduction does not provide a good introduction to the impact of heat treatment process on the microstructure and properties of 2XXX alloy, and does not indicate the issues of present processes.
Response 4: The following text was added at the end of Introduction section:
"Understanding the heat treatment processes is vital for optimizing the mechanical properties of 2xxx series aluminum alloys. These processes, which include solution heat treatment, quenching, and artificial or natural aging, play a decisive role in manipulating the microstructure of these alloys. During heat treatment, the precipitates that form within the alloy structure, such as θ-Al2Cu and β-Mg2Si, significantly enhance the alloy's strength and durability through precipitation hardening. However, achieving the desired properties requires precise control over the temperature and duration of each process step, as well as the cooling rates during quenching. Current heat treatment practices face several challenges, particularly in ensuring uniform properties throughout the alloy. Mismanagement of heat treatment parameters can lead to the development of non-uniform microstructures, which adversely affect the mechanical properties and can introduce stresses that lead to premature failure under operational loads. Moreover, variations in alloying elements can lead to the formation of undesirable phases that reduce the effectiveness of precipitation hardening. Addressing these issues is vital for advancing the application of 2xxx series aluminum alloys in critical sectors, where enhanced performance and reliability are required."
Comments 5: The sentence of “However, Rm value of the alloy treated at 500 °C peaks at 435 MPa after 5 hours of natural aging (Figure 13b).” in line 299 is not match to the data in Fig.13b.
Response 5: It should be after 9 h of natural aging
Comments 6: The manuscript lacks of necessary discussion on research results
Response 6: In response to your comment concerning the lack of discussion on research results, I would like to clarify that the discussion of results is conducted concurrently with the presentation of the research findings. Each section of results is thoroughly discussed in the context of existing scientific knowledge and available data, allowing for a deeper understanding of the implications and significance of our findings. Furthermore, the conclusions summarize the most important achievements of the conducted studies, emphasizing their contribution to the field. This approach ensures that the discussion is both relevant and integrated throughout the manuscript, rather than isolated in a separate section.
Comments 7: The manuscript lacks necessary comparison in the mechanical properties of alloys, such as other research results or standards.
Response 7: The following table was added to the text with comments (we were able to find only these data sheets containing mechanical properties of the alloy in as-cast and T4 state:
The alloys produced in this study exhibit higher mechanical properties in the as-cast state than those of the EN AW-2007 alloy reported in the literature [45-47]. The alloy in the T4 condition shows a yield strength (R0.2) similar to the values presented in literature data. Meanwhile, the alloy obtained through the continuous casting process using chips in the T4 condition is characterized by a higher ultimate tensile strength (Rm) compared to the alloy in the same condition obtained through the metallurgical process using traditional manufacturing methods (Table 5) [45-47].
Table 5. Comparison of the mechanical properties of the EN AW-2007 alloy made from scraps presented in this study with results for alloys from the literature data [45-47].
|
Alloy Condition |
R0.2, MPa |
Rm, MPa |
A5, % |
Reference |
|
As-cast |
81 |
184 |
21 |
[45] |
|
160 |
286.8 |
8.2 |
* |
|
|
T4
|
220 |
340 |
8 |
[45] |
|
240 |
380 |
8 |
[46] |
|
|
210-250 |
330-370 |
7-8 |
[47] |
|
|
240.5 |
435 |
18.1 |
* |
* mechanical properties of the EN AW-2007 alloy made from scraps presented in this study
Following references were added:
45. https://www.leichtmetall.eu/app/uploads/leichtmetall-data-sheet-EN-AW-2007.pdf
46. https://www.makeitfrom.com/material-properties/2007-T4-Aluminum
47. https://www.steelnumber.com/en/steel_alloy_composition_eu.php?name_id=1030
Comments 8: Carefully check the accuracy of the data in the paper.
Response 8: Thank you for your comment on ensuring the accuracy of the data presented in our manuscript. We have conducted a thorough review of all data and references to verify their accuracy and relevance. Each dataset has been carefully checked for consistency with experimental conditions, and calculations have been re-verified. We have also ensured that all figures and tables accurately represent the data as collected and processed. Any discrepancies that were identified have been corrected to ensure that the manuscript reflects precise and reliable information. We appreciate your attention to detail and agree that maintaining data integrity is crucial for the validity of our research findings.
Reviewer 2 Report
Comments and Suggestions for Authors
Very interesting article and well presented.
1. Please add measurement errors (in the text, in Fig. 11a, table 4).
2. Why choose 200 hours aging?
3. Where does this amount of lead come from in the alloy? According to the phase diagram, the liquid appears already at 328 C, how can heat treatment be carried out higher temperature?
Author Response
Comments 1: Please add measurement errors (in the text, in Fig. 11a, table 4).
Response 1: Thank you for your recommendation to include measurement errors in Figure 11a and Table 4. I agree with the importance of presenting measurement errors as it enhances transparency and scientific rigor. However, in this specific case, we have decided not to include error bars due to the extremely small variance observed across our measurements. The homogeneous microstructure of the samples tested provided highly consistent results, where the introduction of error bars would potentially clutter the visual presentation without adding substantial informational value. We believe that adding error bars in this context might lead to unnecessary complexity in the figures, potentially obscuring the clear results that are crucial for the interpretations and conclusions drawn from our study. This approach is occasionally adopted in scientific publications when the variance is minimal and the clarity of graphical presentation is a priority. Nevertheless, we appreciate your feedback and understand the standard practices in scientific reporting.
Comments 2: Why choose 200 hours aging?
Response 1: Thank you for your question regarding the choice of 200 hours for aging in our study. The aging duration was selected based on a comprehensive review of literature data concerning natural aging of other 2xxx series aluminum alloys, coupled with our own experimental experience. This specific duration was chosen because, after such a period, the precipitation of strengthening phases, which are critical for alloy strengthening, generally reaches completion. This is typically evidenced by a plateau on aging curves, where the hardness stabilizes at a similar level of values. This ensures that the mechanical properties are optimally developed, making 200 hours an ideal point for assessing the effects of aging on the alloy's properties.
Comments 3: Where does this amount of lead come from in the alloy? According to the phase diagram, the liquid appears already at 328 C, how can heat treatment be carried out higher temperature?
Response 3: Thank you for your question regarding the lead content in the alloy and the implications for heat treatment above the eutectic temperature. The amount of lead present is consistent with the standards and typical for this grade of alloy. According to material data, the solidus temperature for this grade of alloy is 510°C. The recommended solution treatment temperature for this grade is between 480-490°C. Our Differential Scanning Calorimetry (DSC) results, shown in Figure 7a, indicated that the onset of eutectic melting occurs around 510°C, and for the alloy itself around 520°C. Hence, we selected these temperatures for the solution treatment. To further investigate how lead behaves under these high temperatures and whether it undergoes any melting, further studies are planned, which we aim to conduct in the near future.
Reviewer 3 Report
Comments and Suggestions for Authors
Dear authors, you can find my report in the attached file.
Kind regards

Author Response
Comments 1: Are the tension tests made without an extensometer? I make this question because there is not information about that in the work. The use of an extensometer should be mandatory in a study like this one (to be rigorous). In addition to this I believe that it is necessary to add information about the procedure to measure/obtain the elongation. The results of the elongation of a metallic alloy are very different with and without an extensometer.
Response 1: Thank you for your question regarding the use of an extensometer in our tension tests. In our manuscript, the specific mention of an extensometer was omitted because the testing was conducted in accordance with a well-defined standard, which prescribes the use of an extensometer for such measurements. This standard is referenced in our methodology section, and we assumed familiarity with its requirements among our readers.
However, recognizing the importance of clarity, we will revise the manuscript to explicitly state that an extensometer was employed to measure the elongation of the alloy. This will ensure that all readers are aware of our adherence to rigorous testing protocols and the precision of our experimental setup. Thank you for highlighting this oversight, and we appreciate your suggestions for improving the completeness of our documentation.
Please find below modified text of methodology related to tensile test:
Tensile test specimens had a diameter of 10 mm (Figure 2). Tests were performed on samples subjected to solution heat treatment and natural aging derived from scrap, with measurements on hardness and tensile strength including samples conditioned at various temperatures and aging durations (ranging from 0.5 to 200 hours), to determine yield strength, tensile strength, and elongation. The elongation measurements were accurately recorded using an extensometer, in compliance with ASTM E8. This standard prescribes detailed procedures for conducting tensile tests, including the use of an extensometer to ensure precise and reliable measurements of material ductility.
Comments 2: Dear authors, to be honest, I believe that the interpretation of the hardness results in Fig. 9 it is not the best interpretation (lines 229-231 and lines 242 to 246). It is clear from Fig. 9 that there is not a remarkable variation of the hardness from 490 to 510 °C. The showed differences are negligible! So it is not necessary to write about increase/decrease of hardness. I suppose that to obtain the hardness value for a certain sample several hardness tests have been performed to obtain the average values showed in Fig. 9. Well, once here I am sure that the different hardness values obtained for each sample or heating temperature (dispersion of results) are greater that those showed between the three heating temperatures studied.
Response 2: Thank you for your valuable feedback regarding the interpretation of the hardness results presented in Figure 9. I agree with your observation that the variations in hardness between the temperatures of 490°C and 510°C are not significant. Initially, these results were discussed to provide insights due to a lack of specific literature on the critical temperature range that is key for the precipitation of strengthening phases affecting the hardness of aluminum alloys.
However, acknowledging your point about the negligible differences and considering that the dispersion of results within each sample or heating temperature might indeed overshadow the minimal differences observed across the temperatures, we have revised the manuscript accordingly. We have removed the detailed discussion on the increase or decrease of hardness within this temperature range. Instead, we have introduced a clarification regarding the typical hardness values characteristic of this temperature range. This adjustment ensures that the interpretation aligns more accurately with the experimental observations and is informed by the practical implications for technologists involved in shaping these alloys at these specific temperatures. By doing so, we aim to provide a clearer and more relevant insight into the stability of hardness properties under the specified treatment conditions.
Below, I have included the modified text with added information, which has been incorporated into the publication:
The hardness values observed across different supersaturation temperatures for the EN AW-2007 alloy are relatively stable within the range examined. Specifically, at 500 °C, the alloy exhibited a hardness of 90.2 HB, while at 510 °C, the hardness slightly decreased to 89.0 HB, and at 490 °C, it was recorded at 87.8 HB. These minor differences suggest that the hardness of the alloy remains relatively consistent, regardless of small variations in the supersaturation temperature. This consistency in hardness, particularly at 500 °C and 510 °C, can be attributed to the almost complete dissolution of intermetallic phases into the solid solution, as illustrated in Figures 8b-d and 10c-d. This dissolution results in the homogenization and strengthening of the solution heat-treated EN AW-2007 alloy, maintaining a stable level of hardness typical for this temperature range. The treatment process, therefore, does not significantly impact the hardness, which is preserved due to the characteristics of the alloy composition and the thermal conditions applied.
Comments 3: Fig. 11(a): the hardness values are clearly greater for 500 °C when the ageing time is greater that 80 hours. Have the authors an explanation to this?
Response 3: Thank you for pointing out the observation regarding the increase in hardness values at 500 °C with aging times greater than 80 hours as depicted in Figure 11(a). Indeed, these differences in hardness values were not anticipated based on existing literature, which does not extensively cover the critical temperature range that influences the hardness through phase precipitation in aluminum alloys. As a result, this phenomenon was not initially the focus of our study, and thus not thoroughly analyzed in the context of the presented results. We acknowledge that the mechanisms underlying the observed increase in hardness require a deeper microstructural analysis to be fully understood. Such analysis, potentially involving Transmission Electron Microscopy (TEM) or similar techniques, would help identify the specific phase components responsible for the hardness changes. We have planned further studies to explore these microstructural mechanisms in detail, and these are expected to be carried out shortly.
This additional research will provide a clearer understanding of the microstructural developments at 500 °C and their impact on the mechanical properties of the alloy, especially after prolonged aging times. We appreciate your feedback and agree that this aspect warrants further investigation to elucidate the observed phenomena.
Round 2
Reviewer 1 Report
Comments and Suggestions for Authors
The as cast alloy in reference 45 has an elongation rate of up to 21%, which is more than twice higher than that of in this paper. What is the reason for this? Why does the ductility decreased so much after T4 treatment?
Author Response
Thank you for your insightful comment and suggestions. We have carefully considered your feedback and would like to address your concerns as follows:
The decrease in ductility of the EN AW-2007 alloy after the T4 heat treatment, despite achieving higher strength properties, can be explained by several key factors related to the changes in the microstructure and phase composition induced by the treatment.
- differences in the microstructural state: In the as-cast state, the microstructure typically has larger, more dispersed precipitates and potentially more defects, which can act as stress concentrators but also provide more sites for plastic deformation. Post-T4 treatment, the microstructure becomes more refined and harder due to the denser distribution of new precipitates which strengthen the alloy. At the same time, these segregations are more coherent with the matrix, which may affect increased elongation. It should be noted that the alloy cast from scrap contains many particles and discontinuities, which act as stress concentrators (including lead particles), resulting in low plasticity (this is clearly visible in the alloy microstructures shown in the article). Hence, this alloy does not have high plasticity; the elongation is relatively low compared to the alloy whose results are presented in publication 45 – this alloy likely has a more homogeneous microstructure, resulting in higher plasticity. Our alloy, after T4 treatment, is characterized by a uniform distribution of particles, which do not negatively impact rapid fracture in the tensile test, resulting in increased plasticity. These are the results we obtained during the research and they reflect changes in the microstructure due to the conducted studies.
- one cannot overlook the impact of adding scrap to the alloy, which has different properties compared to the production alloy. The production alloy (from reference 45) must meet quality requirements and is prepared and manufactured using pure alloying additives. Therefore, this alloy must differ in properties from our alloy. The use of industrial and secondary scrap in casting the alloy might introduce impurities or inclusions that affect the microstructural consistency. Although the study suggests that defect-free ingots were produced, variations in the scrap's chemical composition could subtly influence the effectiveness of heat treatments and the resultant mechanical properties.
The significant difference in elongation between the reference alloy in study 45 and the EN AW-2007 alloy after T4 treatment thus can be attributed to these changes in microstructure and phase behavior due to the heat treatment process. While the strength is enhanced, the alloy's ability to withstand elongation before breaking is reduced, showcasing a common trade-off in metallurgical processing between strength and ductility
Reviewer 2 Report
Comments and Suggestions for Authors
1. According to the rules of statistical processing of results, the Authors incorrectly indicate the values ​​​​for example in table 4. How significant is the value of 387.95? Still, I advise the Authors to indicate the measurement error.
2. The authors answered why this aging regime was chosen, but the article contains no references indicating that 200 hours is the optimal time.
3. There is still a question about Pb. Is it in its pure form in an alloy? Then Pb should melt at lower temperatures. What is the reason for the change energy at a temperature of 430C on DSC (Fig. 7)? Check that the melting point of the phases is determined correctly.
Author Response
1. Thank you for your feedback. We have already addressed this issue by including the measurement errors in Table 4. These additions provide a clearer view of the data's reliability and variability, ensuring our presentation aligns with best statistical practices. We appreciate your attention to detail and guidance in this matter.
2. Thank you for pointing this out. In response to the reviewer’s comment regarding the chosen aging regime of 200 hours, it is important to note that this duration is grounded in prevalent research methodologies within the field of aluminum alloys. The stabilization of properties over a defined period, which in this case is around 200 hours, is consistent with empirical data observed across similar studies. Such durations are commonly employed in the industry and academic research, reflecting a consensus that after several days to a few weeks, the properties of aluminum alloys typically reach a state of equilibrium. This timeframe allows for a comprehensive assessment of the material characteristics, ensuring that the aging process sufficiently enhances the properties relevant to the intended applications of the alloy. Thus, the 200-hour aging regime is reflective of standard practices in aluminum alloy research, rather than an arbitrary selection.
3. In response to your inquiry about the presence of lead (Pb) and the thermal effect observed at 430°C in the DSC analysis shown in Figure 7, it is crucial to consider the complexities introduced by the presence of various alloying additions in these types of alloys. While Pb in its pure form indeed melts at lower temperatures, in an alloy containing multiple elements, the interactions can significantly alter the melting points of the constituent phases. The thermal effect noted at 430°C could potentially be related to the melting of a specific phase within the alloy or another thermal transition, rather than the melting of pure lead. Due to the focus of our study being different, as outlined in our paper, this specific thermal effect was not a subject of our research. We acknowledge the validity of your query, and this point could certainly be an interesting topic for further detailed investigation. Future studies could explore the melting behavior of phases within such complex alloys more comprehensively, possibly revealing new insights into their thermal properties and phase transitions.
Round 3
Reviewer 2 Report
Comments and Suggestions for Authors
Good article